# Tocilizumab and steroid treatment in patients with COVID-19 pneumonia

**Malgorzata Mikulska**[1,2]*, **Laura Ambra Nicolini**[2], **Alessio Signori**[3], **Antonio Di Biagio**[1,2], **Chiara Sepulcri**[1], **Chiara Russo**[1], **Silvia Dettori**[1], **Marco Berruti**[1], **Maria Pia Sormani**[3], **Daniele Roberto Giacobbe**[1,2], **Antonio Vena**[2], **Andrea De Maria**[1,2], **Chiara Dentone**[2], **Lucia Taramasso**[2], **Michele Mirabella**[1,2], **Laura Magnasco**[1,2], **Sara Mora**[4], **Emanuele Delfino**[2], **Federica Toscanini**[2], **Elisa Balletto**[1,2], **Anna Ida Alessandrini**[2], **Federico Baldi**[1], **Federica Briano**[1], **Marco Camera**[2], **Ferdinando Dodi**[2], **Antonio Ferrazin**[2], **Laura Labate**[1], **Giovanni Mazzarello**[2], **Rachele Pincino**[1], **Federica Portunato**[2,5], **Stefania Tutino**[1], **Emanuela Barisione**[6], **Bianca Bruzzone**[7], **Andrea Orsi**[7,8], **Eva Schenone**[2], **Nirmala Rosseti**[2], **Elisabetta Sasso**[9], **Giorgio Da Rin**[10], **Paolo Pelosi**[11,12], **Sabrina Beltramini**[9], **Mauro Giacomini**[4], **Giancarlo Icardi**[7,8], **Angelo Gratarola**[13], **Matteo Bassetti**[1,2]

1 Division of Infectious Diseases, Department of Health Sciences (DISSAL), University of Genoa, Genoa, Italy, 2 Division of Infectious Diseases, IRCCS Ospedale Policlinico San Martino, Genoa, Italy, 3 Section of Biostatistics, Department of Health Sciences, University of Genova, Genova, Italy, 4 Department of Informatics, Bioengineering, Robotics and System Engineering, University of Genoa, Genoa, Italy, 5 Infectious Diseases Unit, University of Campania Luigi Vanvitelli, Napoli, Italy, 6 Interventional Pulmonology Unit, IRCCS Ospedale Policlinico San Martino, Genoa, Italy, 7 Hygiene Unit, IRCCS Ospedale Policlinico San Martino, Genova, Italy, 8 Department of Health Sciences, University of Genoa, Genoa, Italy, 9 Pharmacy Complex Unit, IRCCS Ospedale Policlinico San Martino, Genoa, Italy, 10 Medicine Laboratory, IRCCS Ospedale Policlinico San Martino, Genoa, Italy, 11 Department of Surgical Sciences and Integrated Diagnostics, University of Genova, Genova, Italy, 12 Anesthesia and Intensive Care, San Martino Policlinico Hospital, IRCCS for Oncology and Neurosciences, Genoa, Italy, 13 Department of Emergency and Urgency, San Martino Policlinico Hospital, IRCCS for Oncology and Neurosciences, Genoa, Italy

* m.mikulska@unige.it

**Data Availability Statement:** All relevant data are available within the paper and its Supporting Information files.

## Abstract

### Introduction

Coronavirus disease 2019 (COVID-19) can lead to respiratory failure due to severe immune response. Treatment targeting this immune response might be beneficial but there is limited evidence on its efficacy. The aim of this study was to determine if early treatment of patients with COVID-19 pneumonia with tocilizumab and/or steroids was associated with better outcome.

### Methods

This observational single-center study included patients with COVID-19 pneumonia who were not intubated and received either standard of care (SOC, controls) or SOC plus early (within 3 days from hospital admission) anti-inflammatory treatment. SOC consisted of hydroxychloroquine 400mg bid plus, in those admitted before March 24th, also darunavir/ritonavir. Anti-inflammatory treatment consisted of either tocilizumab (8mg/kg intravenously or 162mg subcutaneously) or methylprednisolone 1 mg/kg for 5 days or both. Failure was defined as intubation or death, and the endpoints were failure-free survival (primary

**Funding:** The authors received no specific funding for this work

**Competing interests:** The authors have declared that no competing interests exist

endpoint) and overall survival (secondary) at day 30. Difference between the groups was estimated as Hazard Ratio by a propensity score weighted Cox regression analysis ($HR_{OW}$).

## Results

Overall, 196 adults were included in the analyses. They were mainly male (67.4%), with comorbidities (78.1%) and severe COVID-19 pneumonia (83.7%). Median age was 67.9 years (range, 30–100) and median $PaO_2/FiO_2$ 200 mmHg (IQR 133–289). Among them, 130 received early anti-inflammatory treatment with: tocilizumab (n = 29, 22.3%), methyl-prednisolone (n = 45, 34.6%), or both (n = 56, 43.1%). The adjusted failure-free survival among tocilizumab/methylprednisolone/SOC treated patients vs. SOC was 80.8% (95%CI, 72.8–86.7) vs. 64.1% (95%CI, 51.3–74.0), $HR_{OW}$ 0.48, 95%CI, 0.23–0.99; p = 0.049. The overall survival among tocilizumab/methylprednisolone/SOC patients vs. SOC was 85.9% (95%CI, 80.7–92.6) vs. 71.9% (95%CI, 46–73), $HR_{OW}$ 0.41, 95%CI: 0.19–0.89, p = 0.025.

## Conclusion

Early adjunctive treatment with tocilizumab, methylprednisolone or both may improve outcomes in non-intubated patients with COVID-19 pneumonia.

## Introduction

Pandemic coronavirus disease 2019 (COVID-19) caused by SARS-CoV-2 coronavirus has recently emerged [1]. Although most of the infected patients will remain asymptomatic or develop mild symptoms, up to 20% may develop severe disease with pneumonia and respiratory failure [2]. Oxygen administration is the cornerstone of supportive treatment and is required in approximately 15% of cases, while invasive mechanical ventilation is necessary in up to 5–7% of severe cases [3–5]. Since mortality in patients with invasive ventilation can be very high, halting the progression from moderate to severe respiratory failure should reduce the mortality in COVID-19 [6].

At the beginning of COVID-19 pandemics, based on the experience with previous studies in viral pneumonia, including SARS-CoV and MERS, the use of steroids was discouraged, mainly due to undocumented benefit and fearing potential increase in viral proliferation and side effects [7, 8]. However, with the increasing knowledge about COVID-19, a biphasic model of the disease has been proposed [9, 10]. According to this model, the first phase is caused directly by viral replication, while in the second phase, the symptoms and respiratory failure are due to inflammatory response, and could be treated with agents which reduce inflammation, such as corticosteroids, or inhibitors of pro-inflammatory interleukins and Janus kinase (JAK) [9–11].

Indeed, some real life experiences in COVID-19 patients showed that the use of anti-inflammatory treatments might be beneficial [12]. In fact, short-term steroid therapy was associated with lower mortality in 201 patients with acute respiratory distress syndrome (ARDS) [13]. Additionally, following the data on presence of inflammatory cytokine storm in severe COVID-19, tocilizumab use has been advocated. This monoclonal antibody, which binds to interleukin 6 (IL-6) receptor and blocks the IL-6 mediated inflammatory response, is approved for treatment of rheumatologic disorders and cytokine-release syndrome associated with

Chimeric Antigen Receptor T-cell (CAR-T) administration. It was reported to reduce COVID-19-associated inflammation, and was approved in China for this indication [12, 14].

Based on the first evidences, we formulated the hypothesis of potential benefit of anti-inflammatory treatment, and progressively modified our therapeutic approach to COVID-19. We started using tocilizumab in patients with respiratory failure, and subsequently, we introduced into our protocol early administration of methylprednisolone treatment, followed in more severe cases by the administration of tocilizumab.

We hypothesized that outcomes such as no need for intubation and survival of patients with severe COVID-19 pneumonia in whom tocilizumab and/or methylprednisolone were administered in addition to standard of care (SOC) could be better than in those who received only SOC.

## Patients and methods

### Setting and patients

In this observational single-center study, adult patients admitted to the San Martino University Hospital, Genova, Italy, for COVID-19 pneumonia were included as cases if treated with tocilizumab and/or methylprednisolone, not intubated, not treated with remdesivir and not pregnant. The outcomes of patients treated with tocilizumab/methylprednisolone were compared to data from consecutive patients admitted to our hospital for COVID-19 pneumonia who received only SOC, mainly because they were admitted before the routine use of tocilizumab/methylprednisolone (control group).

All patients provided a verbal informed consent because of isolation precautions for treatment with off label agents according to the local protocol approved by Hospital Authorities, for data collection and for participation in the study, in accordance with national drug agency communication, ver. 2 April 7[th] 2020. The study was carried out in accordance with the principles of the Declaration of Helsinki and approved by the Regional Ethic Committee (N. CER Liguria 114/2020-ID 10420).

### Data collection and definitions

Data were collected from hospital information system by a standard based automatic procedure and stored in an online database with pseudo-anonymization features suitable for secondary use of clinical data [15]. Controls were identified through this prospectively collected database of hospital-admitted COVID-19 patients.

Patients who had any of the following features at the time of, or after, admission were classified as having severe pneumonia: (1) respiratory distress ($\geq$30 breaths per min); or (2) oxygen saturation at rest $\leq$93%; or (3) ratio of partial pressure of arterial oxygen to fractional concentration of oxygen inspired air ($PaO_2/FiO_2$) $\leq$300 mm Hg; or (4) severe disease complications (e.g., respiratory failure, requirement of mechanical ventilation, septic shock, or non-respiratory organ failure) [7, 14, 15].

Systemic inflammation was defined as presence at baseline of one of the following: fever > 38˚C, C-reactive protein (CRP) 10 times above the upper limit of normal (ULN) of 5 mg/dl, ferritin 2 times above the ULN (400 μg/L), or IL-6 10 times above the ULN of 3.4 ng/L. The first results obtained at hospital admission, and in any case not later than within day 3 of admission were considered.

Adverse events possibly or probably related to steroid and tocilizumab treatment, such as neutropenia, anemia, thrombocytopenia, increase in alanine aminotransferase (ALT) levels, (UNL < 40 U/L), microbiologically documented infections and allergic reactions were evaluated for both treated patients and controls. Adverse events were collected up to the last

available follow up from starting of tocilizumab and/or methylprednisolone in treatment group and from hospital admission in the control group. The common terminology criteria for adverse event (CTCAE) version 5.0 was used.

## Standard of care and treatment procedures

The diagnosis of COVID-19 was made with a positive RT-PCR assay performed on nasal swab or broncoalveolar lavage fluid according to World Health Organization interim guidance [16]. Patients received treatment with oral hydroxychloroquine 400mg bid, unless glucose-6-phosphate dehydrogenase deficient. Until March 24[th], darunavir/ritonavir 800/100 qd was also administered [17]. Thereafter, the protocol was amended and darunavir/ritonavir was withdrawn [18]. Short-term antibiotic coverage was prescribed at admission. Low-molecular-weight heparin prophylaxis was administered unless contraindicated. These treatments were defined as SOC.

Since March 11[th], we started adding tocilizumab to SOC in case of severe COVID-19 pneumonia and systemic inflammation. Tocilizumab was administered intravenously at the dose of 8mg/kg (maximum 800mg), with the possibility of repeating the dose after 24 hours if no response was obtained. Due to a temporary shortage of intravenous formulation, the available subcutaneous formulation (162 mg) was administered. Following an internal review of risks and benefits of steroid treatment in patients with severe COVID-19, since March 16[th] methylprednisolone (1mg/kg for 5 days intravenously, then 0.5mg/kg for 5 days) was included in the protocol. Tocilizumab was added in case of systemic inflammation or rapid respiratory function deterioration.

## Statistical analysis

No sample-size calculations were performed. The primary end point was time to failure, defined as intubation and mechanical ventilation or death, whichever occurred first, within 30 days from the hospital admission.

The secondary endpoint was overall survival (OS). Time was calculated from time of hospitalization for the comparison between tocilizumab/methylprednisolone/SOC and SOC patients and from the date of starting anti-inflammation treatment for the comparison among treatment groups.

The landmark analysis was applied in order to minimize the potential immortal time bias that can arise in non-randomized studies and is related to the fact that patients treated after a longer time from admission must have not experienced the event up to that time, and that patients with a very early event (e.g. death) were more likely assigned to the untreated group. This is a conservative analysis which reduced the risk that the treatment choice was motivated by the patient's disease course. Therefore, day 3 from hospital admission was set as a landmark time point: those who died, were intubated or discharged from the hospital before day 3 were excluded, while patients were included in the tocilizumab/methylprednisolone treatment group if the treatment was started within 3 days from hospital admission (see Fig 1).

To minimize baseline differences between treated and untreated patients a propensity score-based analysis was performed. Propensity score (PS) was derived by a logistic regression model including the following baseline variables: age, gender, presence of comorbidities and week of treatment start, ratio of partial pressure of arterial oxygen to fractional concentration of oxygen inspired air ($PaO_2/FiO_2$), non-invasive ventilation (NIV), time from symptoms onset to hospital admission, IL-6, ferritin, C-reactive protein (CRP) and d-dimer serum levels. Positivity assumption of PS was checked after the calculation. For each patient, the overlap weight (OW) based on PS was calculated [19]. To assess the balance of covariate distribution

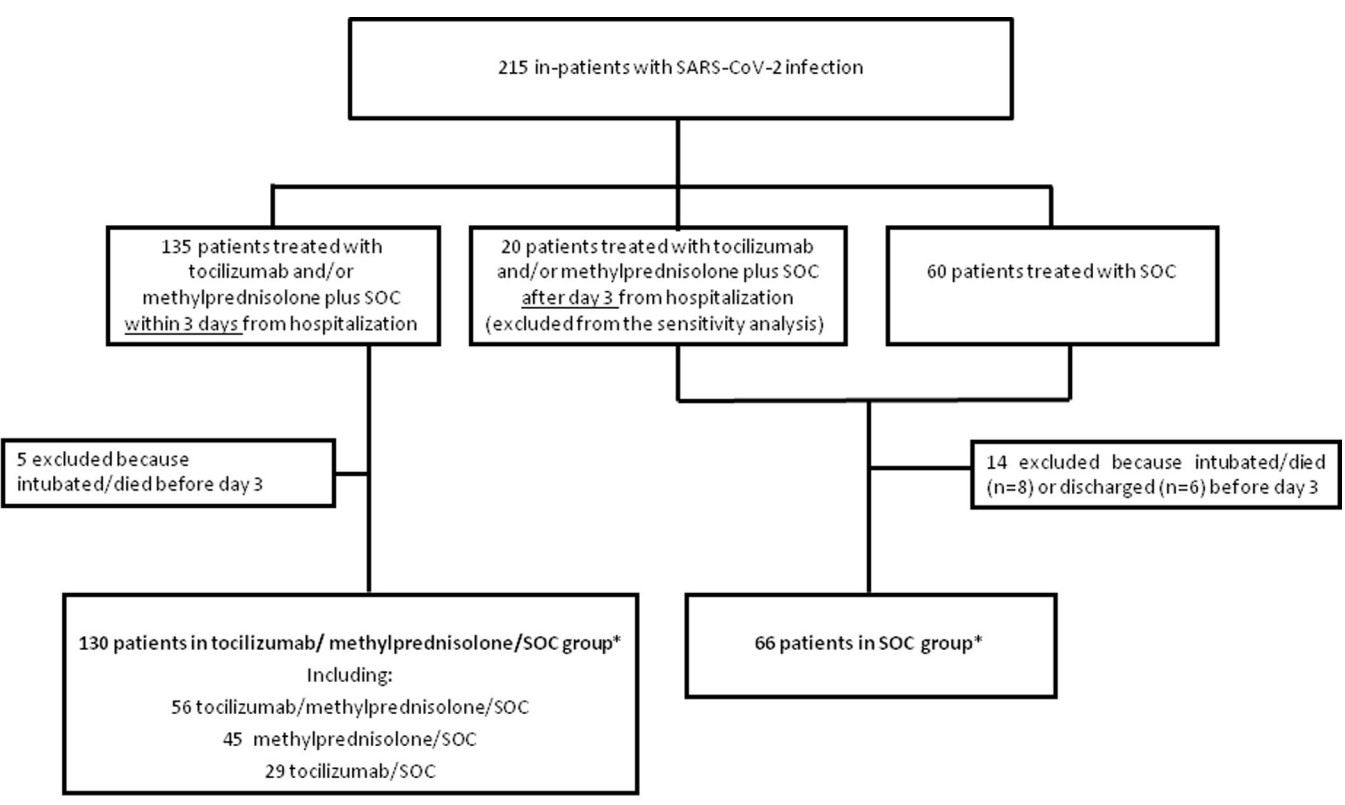

**Fig 1. The flowchart of the included patients.**

between the two groups, Cohen's standardized mean differences were calculated between the two groups in the original samples and after weighting. An absolute value of difference < 0.10 was considered an acceptable balance.

The OW-weighted Cox proportional hazard regression model was used to calculate the adjusted hazard-ratio ($HR_{OW}$) of tocilizumab/methylprednisolone/SOC vs SOC patients.

Weighted cumulative probability of failure or death was calculated by mean of Kaplan-Meier (KM) survival curves.

To define risk factors associated with outcomes and to compare the three treatment groups, adjusted HRs were estimated by a multivariable Cox proportional hazard regression model. The same baseline variables used in the calculation of PS were considered for the multivariable analysis. To avoid overfitting, only those characteristics who showed a p-value ≤ 0.15 at univariable analysis and after inclusion in the multivariable model were considered, with age and gender forced into the model. For a better interpretation and to avoid the influence of outliers on estimation, the IL-6, ferritin, CRP and d-dimer were log-transformed before the analysis due to the highly skewed distribution. All results were reported as HR with 95% confidence interval (95%CI).

A subgroup analysis was performed to assess if the treatment effect of tocilizumab versus methylprednisolone on primary outcome was different between subgroups defined according to categorized baseline variables. An interaction test was used to assess a different treatment effect in subgroups.

The sensitivity analysis was pre-planned, and the comparison between tocilizumab/methylprednisolone/SOC and SOC patients was reassessed excluding from the SOC group the patients that received tocilizumab or methylprednisolone after 3 days from hospitalization. The PS and OW were recalculated.

P was considered significant if $\leq 0.05$. Stata (v.16; StataCorp.) was used for the computation.

## Results

### Patients

Overall, 215 patients were evaluated: 135 (62.8%) received tocilizumab/methylprednisolone within 3 days from hospitalization, 20 (12.9%) were treated after 3 days from hospitalization and were included in the SOC group, and 60 (38.7%) patients received only SOC. After excluding the patients who were discharged or developed a failure event (intubation or death) before the day 3 set for landmark analysis, 196 patients were included (130 tocilizumab/methylprednisolone/SOC and 66 SOC patients) (Fig 1).

Patients were mainly male (67.4%), with a median age of 67.9 years (range, 30–100), and most of them (78.1%) had comorbidities (Table 1). The median $PaO_2/FiO_2$ was 200 mmHg (interquartile range, IQR 133–289), and 164 (83.7%) had severe pneumonia ($PaO_2/FiO_2$ <300mmHg). The median time from the onset of symptoms to anti-inflammatory treatment in 130 patients was 8 days, IQR: 9–15; range 5–23.

### Failure rate

In univariable and multivariable analyses, older age, male gender, higher baseline inflammatory markers, especially IL-6, and $PaO_2/FiO_2 < 100$ mmHg were identified as risk factors for failure (Table 2).

Differences between the two groups were consistently reduced after OW. OW weighted characteristics of two groups of patients are shown in Table 3.

After a median follow-up of 49 days (range 4–70, IQR 30–56), 17 patients were intubated (14/130 in tocilizumab/methylprednisolone/SOC and 3/66 in SOC group) and 36 died (14/130 in tocilizumab/methylprednisolone/SOC and 22/66 in SOC group). Overall, 28 (21.5%) failure events were detected among tocilizumab/methylprednisolone/SOC patients and 25 (37.9%) among SOC patients.

At day 14, the weighted failure-free survival was 86.5% (95%CI, 79.1–91.7) among tocilizumab/methylprednisolone/SOC and 75.8% (95%CI, 63.7–84.1) among SOC patients, while at 30 days (primary endpoint) it was, respectively, 80.8% (95%CI, 72.8–86.7) and 64.1% (95%CI, 51.3–74.0) (Fig 2). The Cox regression analysis adjusted by OW weighted propensity score estimated a significant effect of therapy in reducing the risk of failure ($HR_{OW}$ = 0.48 95%CI, 0.23–0.99; p = 0.049).

### Tocilizumab/methylprednisolone treatment strategies

Within the cohort of 130 treated patients, 45 (34.6%) received methylprednisolone, 29 (22.3%) tocilizumab and 56 (43.1%) combined therapy. Patients in combined treatment group were

**Table 1. Unweighted patients' characteristics stratified for treatment group.**

| Characteristics | All (n = 196) | Tocilizumab/methylprednisolone/SOC (n = 130) | SOC (n = 66) | Standardized difference |
|---|---|---|---|---|
| Age, mean (SD); median (range) | 67.5 (13.7); 67.9 (32–100) | 64.5 (12.4); 65 (32–91) | 73.5 (14.4); 76.2 (39–100) | 0.67 |
| Sex (male/female), n (%) | 132/64 (67.4/32.7) | 91/39 (70/30) | 41/25 (62.1/37.9) | 0.17 |
| Any comorbidity, n (%) | 153 (78.1) | 198 (75.4) | 55 (83.3) | 0.19 |
| Hypertension | 77 (39.3) | 48 (36.9) | 29 (43.9) | |
| Diabetes | 30 (15.3) | 22 (16.9) | 8 (12.1) | |
| Cancer | 22 (11.2) | 15 (11.5) | 7 (10.6) | |
| Obesity | 10 (5.1) | 8 (6.2) | 2 (3.0) | |
| Ischemic heart failure | 22 (11.2) | 11 (8.5) | 11 (16.7) | |
| Hospital admission period | | | | |
| ≤ 11th week of 2020 | 78 (39.8) | 29 (22.3) | 49 (74.2) | 1.21 |
| 12th week of 2020 | 66 (33.7) | 51 (39.2) | 15 (22.7) | 0.36 |
| 13th - 14th week of 2020 | 52 (26.5) | 50 (38.5) | 2 (3.1) | 0.97 |
| Days from symptoms to hospital admission, mean (SD); median (range) | 6.8 (4.1); 7 (0–24) | 7.5 (4.1); 7 (0–24) | 5.6 (4.0); 5 (0–19) | 0.47 |
| IL-6, mean (SD), median (IQR), ng/L | 60.3 (120.3), 36.9 (22–64) | 69.7 (144.8), 40 (27–73) | 41.9 (36.1), 31.5 (16–56) | 0.26 |
| Ferritin, mean (SD), median (IQR), μg/L | 1076 (1060), 762 (452–1424) | 1251.4 (1194.4), 835 (526–1656) | 730.2 (596.7), 588 (316–900) | 0.55 |
| CRP, mean (SD), median (IQR), mg/L | 100.8 (85.3), 81 (40–132) | 112 (91.3), 90 (45–147) | 78.8 (67.2), 65 (30–102) | 0.41 |
| D-dimer, mean (SD), median (IQR), μg/L | 2340 (4091), 1125 (665–1730) | 2319 (4441), 1100 (670–1569) | 2382 (3327), 1173 (651–2806) | 0.02 |
| $PaO_2/FiO_2$, mean (SD), median (IQR), mmHg | 221 (108), 200 (133–289) | 214.8 (97.8) 199 (134–277) | 232.9 (124.3), 208 (130–308) | 0.30 |
| $PaO_2/FiO_2 < 100$ mmHg | 22 (11.2) | 13 (10) | 9 (13.6) | 0.11 |
| $PaO_2/FiO_2 < 200$ mmHg | 93 (47.5) | 65 (50) | 28 (42.4) | 0.15 |
| NIV, n (%) | 73 (37.2) | 64 (49.2) | 9 (13.6) | 0.63 |
| Anti-inflammatory treatment | | | | |
| Steroids only | - | 45 (34.6) | - | - |
| Tocilizumab only | - | 29 (22.3) | - | |
| Combination treatment | - | 56 (43.1) | | |
| Number of doses tocilizumab | | | | |
| 0 | | 45 (34.6) | - | - |
| 1 | - | 74 (56.9) | - | - |
| 2 | - | 11 (8.5) | - | - |
| Route of administration of tocilizumab | | | | |
| Intravenous | - | 49 (57.6) | - | - |
| Subcutaneous | - | 36 (42.4) | - | - |

CRP, C reactive protein; IL-6, interleukin 6; NIV, non invasive ventilation; $PaO_2/FiO_2$, ratio of partial pressure of arterial oxygen to fractional concentration of oxygen inspired air; SOC, standard of care.

younger and with fewer comorbidities but with comparable inflammatory markers, the frequency of low $PaO_2/FiO_2$ and NIV (Table 4).

After a median follow-up of 53 days (range 4–70, interquartile range 33–57), 28 failures were observed: in 14/45 (31.1%) patients receiving methylprednisolone, 6/29 (20.7%) tocilizumab (1 subcutaneously and 5 intravenously) and 8/56 (14.3%) in combined treatment group.

**Table 2. Univariable and multivariable analyses of risk factors for failure.**

| Factors | Univariable | | Multivariable | |
|---|---|---|---|---|
| | HR (95% CI) | p-value | HR (95% CI) | p-value |
| Age (1-year change) | 1.05 (1.03–1.07) | <0.001 | 1.05 (1.01–1.07) | <0.001 |
| Sex (male vs. female) | 1.91 (1.05–3.45) | 0.033 | 1.77 (0.96–3.25) | 0.066 |
| Any comorbidity (Yes vs. No)* | 3.85 (1.55–9.60) | 0.004 | | |
| Treatment period | | 0.17 | | |
| ≤ 11th week of 2020 | 1.00 (ref) | | | |
| 12th week of 2020 | 0.90 (0.52–1.56) | | | |
| 13th - 14th week of 2020 | 0.54 (0.27–1.07) | | | |
| Time from symptoms (>7 vs ≤ 7 days) | 1.01 (0.61–1.67) | 0.97 | | |
| IL-6 (1-unit on log-scale) | 1.84 (1.45–2.33) | <0.001 | 1.70 (1.30–2.23) | <0.001 |
| Ferritin (1-unit on log-scale) | 1.10 (0.84–1.44) | 0.47 | | |
| CRP (1-unit on log-scale) | 1.61 (1.21–2.16) | 0.001 | | |
| D-dimer (1-unit on log scale) | 1.50 (1.20–1.87) | <0.001 | | |
| $PaO_2/FiO_2$ <100 vs ≥100 | 3.12 (1.79–5.45) | <0.001 | 1.95 (1.10–3.46) | 0.023 |
| NIV | 1.28 (0.78–2.09) | 0.33 | | |

CRP, C reactive protein; HR: hazard-ratio; IL-6, interleukin 6; NIV, non invasive ventilation; $PaO_2/FiO_2$, ratio of partial pressure of arterial oxygen to fractional concentration of oxygen inspired air.

*None of the following single comorbidities resulted significant in multivariable analysis: hypertension, diabetes, cancer, obesity, ischemic heart failure, and only ischemic hear disease was significant in univariate analysis.

At 14 days of follow-up from treatment start, the failure-free survival (Fig 3) was 80% (95% CI, 65.1–89.1) in methylprednisolone group, 79.3% (95%CI, 59.6–90.1) in tocilizumab group and 87.5% (95 CI, 75.6–93.8) in combined therapy group. No significant differences between

**Table 3. OW weighted patients' characteristics stratified for treatment group.**

| Characteristics | Tocilizumab/methilprednisolone/SOC (n = 130) | SOC (n = 66) | Standardized weighted difference |
|---|---|---|---|
| Age, mean (SD); median (IQR) | 68.4 (11.7); 68.3 (57.4–77.3) | 68.4 (13); 68 (59.2–76.9) | 0.000 |
| Sex (male/female), n (%) | 87/43 (67/33) | 44/22 (67/33) | 0.000 |
| Any comorbidity, n (%) | 102 (78.8) | 52 (78.8) | 0.000 |
| Treatment period | | | |
| ≤ 11th week of 2020 | 68 (52.1) | 35 (52.1) | 0.000 |
| 12th week of 2020 | 52 (40.0) | 26 (40) | 0.000 |
| 13th - 14th week of 2020 | 10 (7.9) | 5 (7.9) | 0.000 |
| Days from symptoms to hospital admission, mean (SD); median (IQR) | 7 (3.6); 7 (4–9) | 6.8 (4.2); 7 (3–9) | 0.071 |
| Il-6, mean (SD); median (IQR), ng/L | 45.2 (55.7); 37 (26–51) | 45.2 (36.8); 32.4 (17–64) | 0.000 |
| Ferritin, median (IQR), µg/L | 901 (671); 762 (491–1095) | 901 (727); 706 (479–995) | 0.000 |
| PCR, median (IQR), mg/L | 90.6 (65.3); 73 (43–124) | 90.6 (83.3); 63.1 (30–131) | 0.000 |
| D-dimer, median (IQR) | 1764 (3110); 1080 (680–1500) | 1764 (2372); 1030 (651–1691) | 0.000 |
| $PaO_2/FiO_2$, mean (SD); median (IQR), µg/L | 211.6 (96.1); 196 (144–280) | 211.6 (118); 180 (123–291) | 0.000 |
| NIV, n (%) | 53 (41.1) | 27 (41.1) | 0.000 |

NIV, non invasive ventilation; OW: overlap weights; $PaO_2/FiO_2$, ratio of partial pressure of arterial oxygen to fractional concentration of oxygen inspired air; SOC, standard of care.

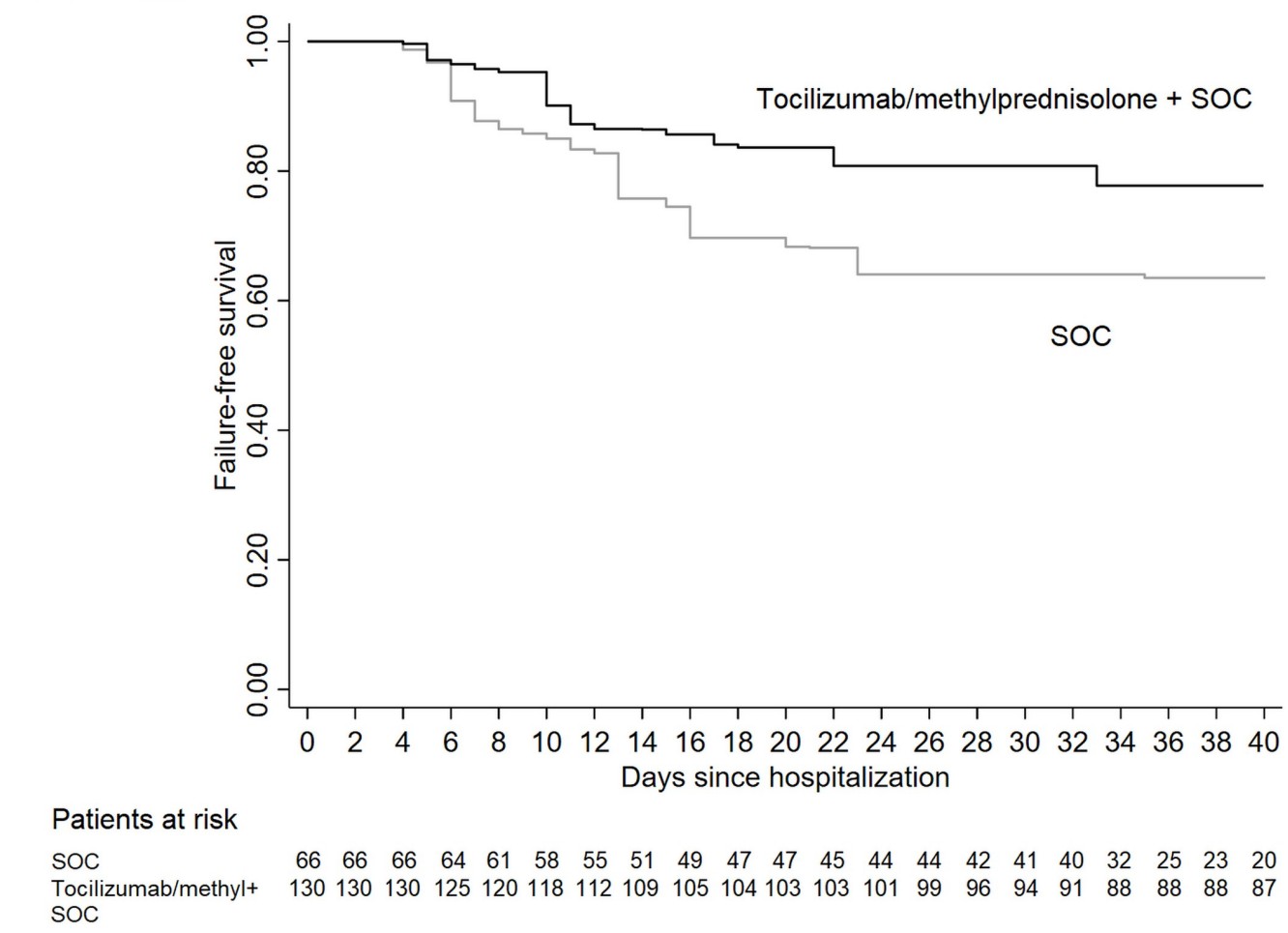

**Fig 2. Adjusted failure-free (intubation or death) survival in patients who received tocilizumab and/or methylprednisolone and Standard Of Care (SOC) vs. SOC.**

the three treatment groups were identified in a multivariable analysis adjusted for the baseline risk factors (p for heterogeneity among treatment groups = 0.45) (Table 5).

## Overall Survival (OS)

A total of 45 deaths, 36 without a previous intubation, were registered during the follow-up: 23 (34.8%) were in the SOC and 22 (16.9%) in tocilizumab/methylprednisolone/SOC group.

At day 14 of follow-up, the OS was 92.7% (95%CI, 86.4–96.1) in tocilizumab/methylpred-nisolone/SOC group and 78.2% (95%CI, 67.0–85.6) in SOC (Fig 4). At 30 days OS was, respectively, 85.9% (95%CI, 80.7–92.6) and 71.9% (95%CI, 59.9–80.8), with a significant better survival in tocilizumab/methylprednisolone/SOC patients in OW analysis ($HR_{OW}$ = 0.41, 95% CI: 0.19–0.89, p = 0.025).

When considering only patients receiving tocilizumab/methylprednisolone/SOC, death occurred in 13/45 (28.9%) patients receiving methylprednisolone, 4/29 (13.8%) tocilizumab and 5/56 (8.9%) in those receiving combined therapy. At 30 days the OS was, 79.5% (95%CI, 64.2–88.8), 85.8% (95%CI, 66.5–94.4) and 90.9% (95%CI, 79.6–96.1) in patients treated, respectively, with methylprednisolone, tocilizumab and combined therapy (adjusted p for het-erogeneity among treatments: 0.55).

**Table 4. Clinical and demographic characteristics in treated patients included in outcome analyses.**

| Characteristics | Methylprednisolone (n = 45) | Tocilizumab (n = 29) | Combined (n = 56) |
|---|---|---|---|
| Age, mean (SD); median (range) | 67.5 (13.9); 68.2 (37–91) | 65.9 (10.2); 67.4 (42–80) | 61 (11.7); 63 (32–85) |
| Sex (male/female), n (%) | 32/13 (71.1/28.9) | 24/5 (82.8/17.2) | 35/21 (62.5/37.5) |
| Any comorbidity, n (%) | 36 (80) | 23 (79.3) | 39 (69.6) |
| Treatment period | | | |
| $\leq$ 11th week of 2020 | 4 (8.9) | 22 (75.8) | 3 (5.4) |
| 12th week of 2020 | 30 (66.7) | 6 (20.7) | 15 (26.8) |
| 13th - 14th week of 2020 | 11 (24.4) | 1 (3.5) | 38 (67.9) |
| Days from symptoms to hospital admission, mean (SD); median (range) | 7.2 (4.7); 7 (0–21) | 7.7 (3.5); 8 (1–15) | 7.6 (3.8); 7 (2–24) |
| IL-6, median (IQR), ng/L | 40 (22–68) | 45 (33–88) | 37 (27–63) |
| Ferritin, median (IQR), µg/L | 741 (380–1087) | 1422 (845–1846) | 760 (444–1858) |
| CRP, median (IQR), mg/L | 82 (43–124) | 121 (52–218) | 88 (44–153) |
| D-dimer, median (IQR), µg/L | 1080 (730–1600) | 1280 (950–2500) | 1055 (618–1411) |
| $PaO_2/FiO_2$, median (IQR), mmHg | 201 (156–311) | 203 (144–280) | 190 (125–254) |
| $PaO_2/FiO_2 < 100$ mmHg | 6 (13.3) | 1 (3.5) | 6 (10.7) |
| $PaO_2/FiO_2 < 200$ mmHg | 22 (48.9) | 13 (44.8) | 30 (53.6) |
| NIV, n (%) | 20 (44.4) | 13 (44.8) | 31 (55.4) |
| Number of doses tocilizumab | | | |
| 0 | | - | - |
| 1 | | 22 (75.9) | 54 (96.4) |
| 2 | | 7 (24.1) | 2 (3.6) |
| Route of administration of tocilizumab | | | |
| Intravenous | | 18 (62.1) | 31 (55.4) |
| Subcutaneous | | 11 (37.9) | 25 (44.6) |

CRP, C reactive protein; IL-6, interleukin 6; NIV, non invasive ventilation; $PaO_2/FiO_2$, ratio of partial pressure of arterial oxygen to fractional concentration of oxygen inspired air.

## Subgroup analyses

It was not possible to identify subgroups, based on demographic or clinical characteristics, with different benefits of tocilizumab versus methylprednisolone or tocilizumab versus combination therapy.

## Sensitivity analyses

As a sensitivity analysis, we run the comparison between tocilizumab/methylprednisolone/SOC and SOC patients excluding the 20 patients that received tocilizumab/methylprednisolone treatment after 3 days from hospital admission (6 methylprednisolone, 8 tocilizumab and 6 both), and experienced 5 (25%) failures. The weighted failure-free survival in the SOC group was now 72.5% (95%CI, 58.8–81.5) at 14 days and 67.1% (95%CI, 52.8–77.9) after 30 days. The OW weighted difference between tocilizumab/methylprednisolone/SOC and SOC patients was amplified ($HR_{OW}$ = 0.39; 95%CI, 0.17–0.89; p = 0.026).

## Side effects

Among 155 patients who received tocilizumab/methylprednisolone/SOC at any time, 106 (68%) developed ALT increase (grade 1–2: 98/155, 63% and grade 3: 8/155, 5%): 32.4% of those who received methylprednisolone/SOC and 77% of those who received tocilizumab/SOC +/- methylprednisolone. Microbiologically documented infections were recorded in 12

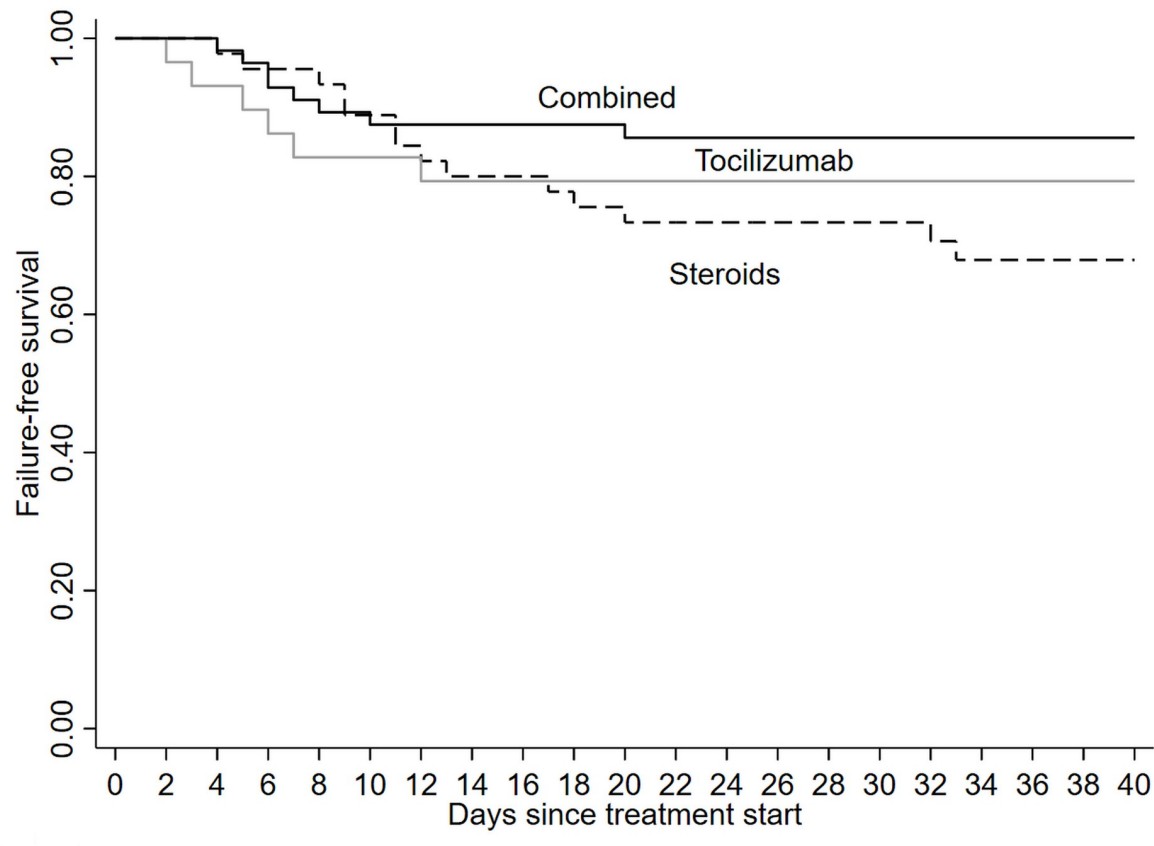

**Fig 3. Adjusted failure-free survival in patients receiving anti-inflammatory treatment: Tocilizumab only, methylprednisolone only, and combined treatment.**

(7.7%) patients: 10 bloodstream infections, 2 ventilator-associated pneumonias and 2 urinary tract infections. In tocilizumab/methylprednisolone/SOC group, grade 4 transient neutropenia and grade 2 maculopapular rash occurred in 1 patient each.

Among 60 SOC patients, grade 1–2 ALT increase occurred in 5 (8%) and 1 patient developed urinary tract infection.

There were no cases of significant decrease of hemoglobin or platelet levels.

## Discussion

In this observational study in non-intubated patients with mainly severe COVID-19 pneumonia, the early addition of tocilizumab and/or methylprednisolone to SOC resulted in adjusted failure-free survival of 86.5% and 80.8% at day 14 and 30, which was, respectively, 10.7% and 16.7% higher than in SOC patients.

Even though only the minority of patients develop the severe form of COVID-19 (14% in the Chinese cohort of 72.314 patients), the outcomes in this group are poor, with 24.9% rate of failure (intubation or death) reported in 173 patients with severe disease [2, 20]. The observation

**Table 5. Univariable and multivariable analyses for risk factors of failure endpoint in patients treated with tocilizumab and/or methylprednisolone and Standard Of Care (SOC).**

| Factors | Univariable | | Multivariable | |
|---|---|---|---|---|
| | HR (95% CI) | p-value | HR (95% CI) | p-value |
| Treatment | | 0.17 | | 0.45 |
| Methylprednisolone | 1.00 (ref) | | 1.00 (ref) | |
| Tocilizumab | 0.69 (0.27–1.81) | | 0.65 (0.23–1.82) | |
| Combined | 0.45 (0.19–1.06) | | 0.59 (0.24–1.43) | |
| Age (1-year change) | 1.05 (1.02–1.09) | 0.002 | 1.05 (1.01–1.09) | 0.012 |
| Sex (male vs. female) | 1.68 (0.68–4.14) | 0.26 | 1.54 (0.61–3.91) | 0.37 |
| Any comorbidity (Yes vs. No) | 4.76 (1.13–20.04) | 0.034 | - | |
| Treatment period | | 0.46 | | |
| $\leq 11^{th}$ week of 2020 | 1.00 (ref) | | - | |
| $12^{th}$ week of 2020 | 1.03 (0.41–2.59) | | - | |
| $13^{th}$ - $14^{th}$ week of 2020 | 0.62 (0.22–1.70) | | - | |
| Time from symptoms (>7 vs $\leq$ 7 days) | 0.93 (0.44–1.98) | 0.86 | - | |
| Il-6 (1-unit on log-scale) | 1.93 (1.38–2.70) | <0.001 | 1.87 (1.29–2.71) | 0.001 |
| Ferritin (1-unit on log-scale) | 1.20 (0.78–1.83) | 0.41 | - | |
| CRP (1-unit on log-scale) | 1.62 (1.02–2.57) | 0.04 | - | |
| D-dimer (1-unit on log scale) | 1.62 (1.19–2.21) | 0.002 | - | |
| PaO2/FiO2 <100 vs $\geq$100 mmHg | 2.70 (1.09–6.66) | 0.032 | 1.88 (0.72–4.70) | 0.14 |
| NIV | 1.99 (0.90–4.40) | 0.090 | - | |

CRP, C reactive protein; HR: hazard-ratio; IL-6, interleukin 6; NIV, non invasive ventilation; $PaO_2/FiO_2$, ratio of partial pressure of arterial oxygen to fractional concentration of oxygen inspired air.

that COVID-19-associated respiratory failure can be caused by cytokine storm rather than viral progression is the rationale for administering anti-inflammatory treatments, including tocilizumab [9, 10, 21]. The initial studies on tocilizumab in COVID-19 reported a clinical benefit in retrospective cohorts of 21 and 15 patients with moderate to critical COVID-19 pneumonia, in whom steroids were also administered [12, 22]. However, none of studies reported and evaluated the impact of steroid co-administration, nor included a control group which did not receive tocilizumab. Subsequently these treatments were recommended by Chinese Diagnosis and Treatment Protocol for Novel Coronavirus Pneumonia (version 6 and 7) and experience from larger cohorts in Europe have been published [14, 23, 24]. However, considering the rapid widespread increase in severe cases of COVID-19 worldwide, the availability and the cost of anti-IL-6 treatment might limit its use. Therefore, at the peak of COVID-19 epidemics in our city, we implemented the early use of corticosteroids and the use of subcutaneous tocilizumab if intravenous formulation was not readily available. We acknowledge that the benefit of subcutaneous formulation might be lower and slower than in case of intravenous drug, but no standard intravenous dose for COVID-19 has been established, as one study used 400 mg and the other the range of doses from 80 mg to 600 mg, irrespective of the patients' weight [12, 22].

The first study that reported the impact of steroid treatment in COVID-19, showed that it was administered to 30.8% of patients, mainly in case of more severe disease, and was associated with a significant reduction of the risk of death in patients with ARDS (HR = 0.38) [13]. While other cohorts reported steroid use in approximately 44% of severely ill patients, they did not analyze their influence on outcome [20, 25]. Subsequently, an observational study reported a benefit of early administration of steroid therapy on composite outcome of ICU admission, mechanical ventilation or death at 14 days (34.9% vs. 54.3%) and overall survival (86.4% vs.

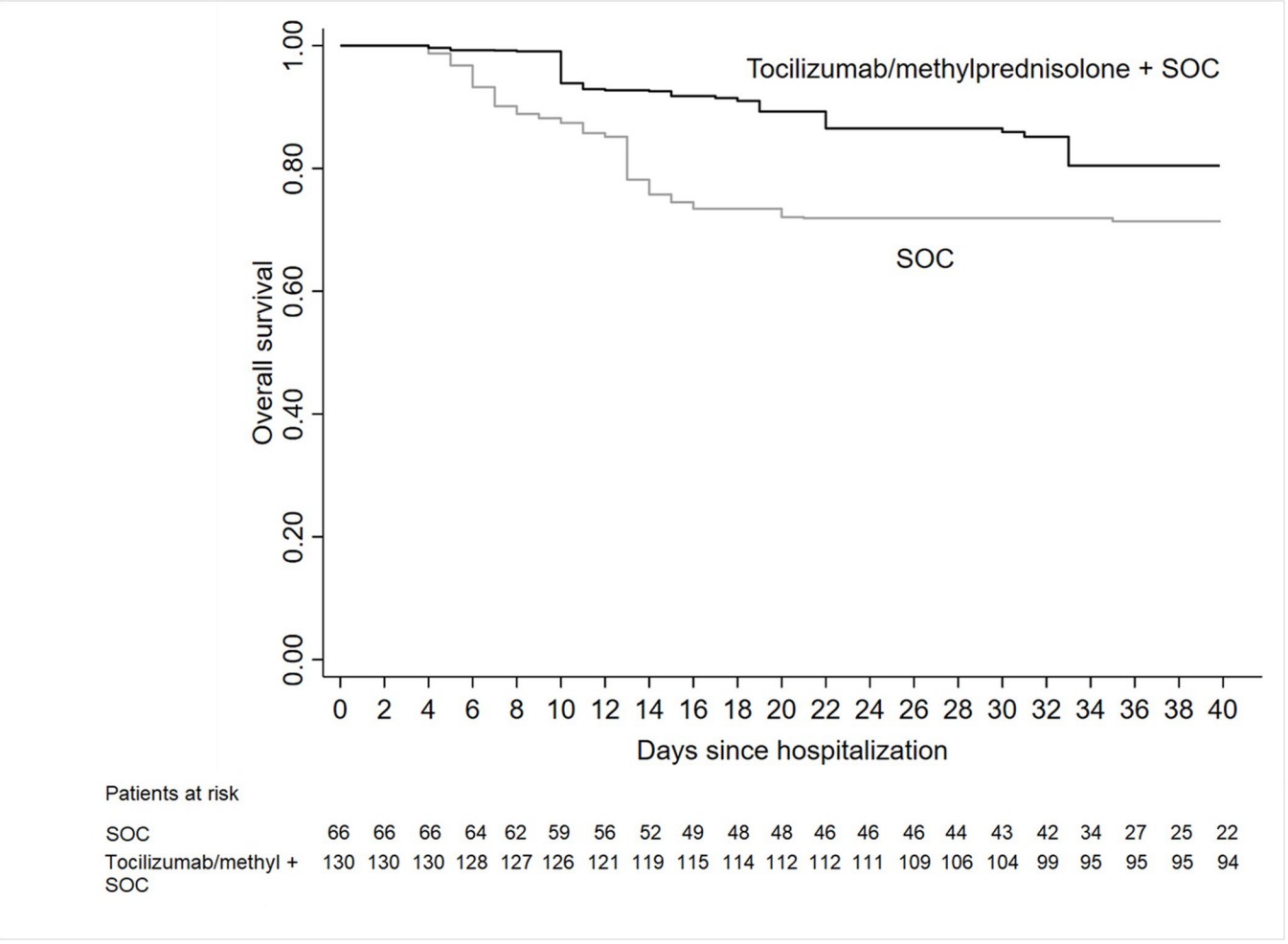

**Fig 4. Adjusted overall survival in patients who received tocilizumab and/or methylprednisolone and Standard Of Care (SOC) vs. SOC.**

73.7%) [26]. In addition to these data, we were able to demonstrate that, after minimizing as much as possible the differences between the groups through OW adjustment, the outcome of patients was better in case of early treatment with tocilizumab and/or methylprednisolone. Indeed, in SOC group the rate of failure at day 14 of 24.2% was very similar to what reported in other cohorts with severe pneumonia (24.9%), while it was reduced to 13.5% in our tocilizumab/methylprednisolone/SOC group [20]. Our data show that this benefit was also present with month-long follow up (overall $HR_{ow}$ of 0.48), which is important in establishing long term prognosis of these patients. Moreover, the benefit of early tocilizumab/methylprednisolone was also noted on overall survival, both at 14 and 30 days (respectively, 92.7% vs. 78.2% and 85.9% vs. 71.9%). Compared to the study with steroid use only, the 14-day survival was higher in our cohort, providing background for the hypothesis that combined tocilizumab/steroid treatment might be warranted [26]. Finally, preliminary results of the RECOVERY trial reported higher survival rate in patients with severe COVID-19 pneumonia treated with dexamethasone, although the detailed results are not available yet. Interestingly, our observational study documented that these patients were treated at a median time of 8 days after the onset of symptoms, which is compatible with the timing of cytokine storm, and therefore might be optimal for the effect of anti-inflammatory treatment.

Consistent with other studies, we identified older age, high IL-6 levels and poor respiratory function as independent predictors of failure, with possible impact also of CRP and d-dimer levels [27]. Possibly due to a limited sample size, we were unable to document which of three treatment groups provided most benefit, and if there were predictors of better response to tocilizumab/methylprednisolone compared to methylprednisolone alone in any subset of patients. However, the rate of failure-free survival was the highest in the combination treatment group. In addition, based on our sensitivity analyses, adding the anti-inflammatory treatment later after hospital admission might still provide some clinical benefit. In fact, including in the SOC group patients who received anti-inflammatory treatment later during the infectious course possibly reduced the difference between the study arms, supporting the overall benefit of an early anti-inflammatory treatment.

The limitations of this study include the non-randomized design, yet the inclusion of consecutive patients using the same SOC but not treated with tocilizumab or methylprednisolone, and adjustment for the outcome-associated variables, allowed to note the improvement in patient outcomes. Nonetheless, it is possible that some benefit observed was partially due to general improvements in patient clinical care that occur with time. Additionally, this being a single-center experience might limit the applicability to other settings, since our hospital managed to rapidly increase the capacity for hospitalization and ventilation support, potentially improving general patient care. However, the adjustment for the differences between patient groups through propensity score and conservative approach with the use of landmark analysis were directly at minimizing the risk associated with an absence of randomization. Finally, we believe that the rate of failure observed in this study of 7.3% at 14 days in those with severe COVID-19 treated with SOC and tocilizumab/methylprednisolone might help to better define the expected rate of response and calculate the number of patients needed to include in the studies assessing various treatment options.

## Conclusion

In conclusion, the negative impact of immune response in COVID-19 might be mitigated by early administration of anti-inflammatory therapy with tocilizumab, methylprednisolone or both. Randomized studies are warranted to establish the best treatment options, their timing and limitations.

## Supporting information

**S1 Dataset.**
(XLS)

## Acknowledgments

We would like to thank all the patients and the hospital staff, with particular mention of Mrs Enrica Lombardi, who helped us to get through these difficult weeks.

This study was supported by the efforts of all members of GECOVID group.

## Author Contributions

**Conceptualization:** Malgorzata Mikulska, Antonio Di Biagio, Chiara Dentone, Matteo Bassetti.

**Data curation:** Malgorzata Mikulska, Laura Ambra Nicolini, Chiara Sepulcri, Chiara Russo, Silvia Dettori, Marco Berruti, Antonio Vena, Michele Mirabella, Laura Magnasco, Sara

Mora, Elisa Balletto, Federico Baldi, Federica Briano, Marco Camera, Laura Labate, Rachele Pincino, Federica Portunato, Stefania Tutino, Elisabetta Sasso.

**Formal analysis:** Malgorzata Mikulska, Alessio Signori, Maria Pia Sormani.

**Investigation:** Malgorzata Mikulska, Laura Ambra Nicolini, Emanuele Delfino, Federica Toscanini, Anna Ida Alessandrini, Ferdinando Dodi, Antonio Ferrazin, Giovanni Mazzarello, Emanuela Barisione, Bianca Bruzzone, Andrea Orsi, Eva Schenone, Nirmala Rosseti.

**Methodology:** Malgorzata Mikulska, Laura Ambra Nicolini, Alessio Signori, Maria Pia Sormani, Andrea De Maria, Matteo Bassetti.

**Software:** Sara Mora, Mauro Giacomini.

**Supervision:** Antonio Di Biagio, Daniele Roberto Giacobbe, Emanuele Delfino, Giorgio Da Rin, Paolo Pelosi, Sabrina Beltramini, Giancarlo Icardi, Angelo Gratarola, Matteo Bassetti.

**Validation:** Malgorzata Mikulska, Paolo Pelosi, Matteo Bassetti.

**Visualization:** Malgorzata Mikulska, Antonio Di Biagio, Daniele Roberto Giacobbe, Antonio Vena, Chiara Dentone, Lucia Taramasso, Emanuele Delfino, Federica Toscanini, Anna Ida Alessandrini, Ferdinando Dodi, Antonio Ferrazin, Paolo Pelosi, Matteo Bassetti.

**Writing – original draft:** Malgorzata Mikulska, Laura Ambra Nicolini.

**Writing – review & editing:** Malgorzata Mikulska, Laura Ambra Nicolini, Daniele Roberto Giacobbe, Antonio Vena, Andrea De Maria, Lucia Taramasso, Laura Magnasco, Paolo Pelosi, Matteo Bassetti.

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
