## [Decision Letter · Decision Letter 0]

5 Aug 2020

Tocilizumab and steroid treatment in patients with COVID-19 pneumonia.

PONE-D-20-19437

Dear Dr. Mikulska,

We’re pleased to inform you that your manuscript has been judged scientifically suitable for publication and will be formally accepted for publication once it meets all outstanding technical requirements.

Kind regards,

Muhammad Adrish

Academic Editor

PLOS ONE

1.Please amend your current ethics statement to address the following concerns:  

a) Did participants provide their written or verbal informed consent to participate in this study?

Reviewers' comments:

Reviewer's Responses to Questions

**Comments to the Author**

1. Is the manuscript technically sound, and do the data support the conclusions?

Reviewer #1: Yes

Reviewer #2: Yes

2. Has the statistical analysis been performed appropriately and rigorously? 

Reviewer #1: Yes

Reviewer #2: Yes

3. Have the authors made all data underlying the findings in their manuscript fully available?

Reviewer #1: Yes

Reviewer #2: No

4. Is the manuscript presented in an intelligible fashion and written in standard English?

Reviewer #1: Yes

Reviewer #2: Yes

5. Review Comments to the Author

Reviewer #1: this study provide important experience in treatment and saving the patients with COVID- 19 by using the biological therapy in form of interlucin-6 inhibitor (tocilizumab) and anti-inflammatory and immune-suprsive in form of coticosteroid.

Reviewer #2: The manuscript titled “Tocilizumab and steroid treatment in patients with COVID-19 pneumonia” by Mikulska, M. et. al reports the outcomes of treating patients with Covid-19 pneumonia with antinflammatory therapy including the corticosteroid Methylprednisolone with or without the IL-6 inhibitor tocilizumab. This is a single center observational study and was not a randomized placebo-controlled trial. However the study parameters were selected appropriately and the statistical analyses conducted robustly to minimize the effects of multiple biases and to obtain medically valid data. The results seems to indicate that the use of tocilizumab with methylprednisolone seems to be advantageous in improving the outcomes in patients with pneumonia associated with Covid-19. More multi-centric studies are necessary to make this treatment universally applicable. This reviewer has no major concerns with this manuscript.

6. PLOS authors have the option to publish the peer review history of their article (what does this mean?). If published, this will include your full peer review and any attached files.

Reviewer #1: **Yes: **Amal Bakry Abdul sattar

Reviewer #2: No

---

## [Editor Report · Acceptance letter]

12 Aug 2020

PONE-D-20-19437 

Tocilizumab and steroid treatment in patients with COVID-19 pneumonia. 

Dear Dr. Mikulska:

I'm pleased to inform you that your manuscript has been deemed suitable for publication in PLOS ONE. Congratulations! Your manuscript is now with our production department. 

Kind regards, 

on behalf of

Dr. Muhammad Adrish 

Academic Editor

PLOS ONE